# Macrophage Polarization: An Important Candidate Regulator for Lung Diseases

**DOI:** 10.3390/molecules28052379

**Published:** 2023-03-04

**Authors:** Lishuang Deng, Zhijie Jian, Tong Xu, Fengqin Li, Huidan Deng, Yuancheng Zhou, Siyuan Lai, Zhiwen Xu, Ling Zhu

**Affiliations:** 1College of Veterinary Medicine, Sichuan Agricultural University, Chengdu 625014, China; 2College of Animal Science, Xichang University, Xichang 615000, China; 3Livestock and Poultry Biological Products Key Laboratory of Sichuan Province, Sichuan Animal Science Academy, Chengdu 625014, China; 4Key Laboratory of Animal Disease and Human Health of Sichuan Province, Chengdu 625014, China

**Keywords:** macrophage, polarization, M1/M2, signaling pathway, lung disease

## Abstract

Macrophages are crucial components of the immune system and play a critical role in the initial defense against pathogens. They are highly heterogeneous and plastic and can be polarized into classically activated macrophages (M1) or selectively activated macrophages (M2) in response to local microenvironments. Macrophage polarization involves the regulation of multiple signaling pathways and transcription factors. Here, we focused on the origin of macrophages, the phenotype and polarization of macrophages, as well as the signaling pathways associated with macrophage polarization. We also highlighted the role of macrophage polarization in lung diseases. We intend to enhance the understanding of the functions and immunomodulatory features of macrophages. Based on our review, we believe that targeting macrophage phenotypes is a viable and promising strategy for treating lung diseases.

## 1. Introduction

Macrophages are a crucial cell type in the innate immune system, possessing complex physiological functions and a wide distribution throughout the body. They not only serve as the first line of defense against pathogens in the host’s innate immune response, but also play a vital role in maintaining organismal homeostasis by participating in waste removal and tissue repair [1]. Macrophages are an extremely heterogeneous population with a high degree of plasticity [2,3]. Depending on the stimulus or local microenvironment, macrophages can polarize into multiple phenotypes, mainly consisting of classically activated macrophages (M1) and selectively activated macrophages (M2) [4,5,6,7].

In the lungs, macrophages constitute the most abundant immune cell population, comprising approximately 70% [8,9]. They are associated with the pathophysiology of a diverse array of lung diseases, including acute lung injury (ALI), chronic obstructive pulmonary disease (COPD), asthma, tuberculosis, and lung cancer [10,11,12,13,14]. During the pathology of various diseases, macrophages are critical in regulating immune homeostasis by regulating the M1/M2 polarization. Macrophage polarization has garnered significant attention as a research hotspot in recent years, and its regulation holds promise as a novel therapeutic strategy for disease treatment [15]. In the current review, we discuss the origin of macrophages, the phenotype and polarization of macrophages, the signaling pathways related to macrophage polarization, and the role of macrophage polarization in lung diseases. We aim to enhance the understanding of the functions and immunomodulatory features of macrophages.

## 2. Origin of Macrophages

Initially, it was widely believed that tissue macrophages in health and disease were derived from circulating monocytes [16,17]. However, the discovery of resident tissue macrophages in multiple organ systems that originate from the yolk sac during embryonic development has led to a dramatic shift in researchers’ understanding of macrophage origin [9,18]. At present, a large body of evidence demonstrates that tissue macrophages have a dual origin [19,20]. Macrophages can either differentiate from circulating monocytes derived from bone marrow stem cells or derive from primitive macrophages originating from the embryonic yolk sac and fetal liver. These primitive macrophages take up residence during the prenatal embryonic development, where they proliferate to form a resident tissue macrophage population [21] (Figure 1).

Macrophages are distributed in almost all organs and tissues throughout the body, where they play a diverse range of physiological roles during growth and development [22,23,24]. In some organs and tissues, macrophages are given unique names, such as microglia in the brain, alveolar macrophages in the lungs and Kupffer cells in the liver [23,25,26]. Alveolar macrophages and Kupffer cells derive from fetal liver mononuclear cells, while microglia derive from yolk sac erythromyeloid progenitors in early embryonic development. The origin and diversity of macrophages in different tissues are significant for both homeostatic functions and pathology. In general, embryonic-derived macrophages play a crucial role in maintaining tissue homeostasis, and macrophages derived from bone marrow monocytes are associated with host defense responses and pathological signaling [1,16,20]. However, each organ and its surrounding environment influence the function and properties of macrophages [27,28].

## 3. Phenotype and Polarization of Macrophages

Mature macrophages exhibit high plasticity and heterogeneity in phenotype and function, enabling them to respond and adapt to various environmental signals. Based on the concept of type-1/type-2 helper T (h-) cell polarization, phenotypically polarized macrophages are defined according to two main activation states, namely M1 and M2 macrophages (Figure 2). The nomenclature of M1 and M2 macrophages was first proposed by Mills et al. and has been widely used to define the major subsets of macrophages for a long time [5]. Upon sensing changes in the local tissue microenvironment, macrophages can activate different signal cascades, leading to the polarization into different phenotypes, or even switching from one phenotype to another. Many physiological and pathological states of the body are closely related to the proportion of the two macrophage phenotypes [7,15,29].

### 3.1. M1 Macrophages

M1 macrophages, also known as “classically activated (proinflammatory) macrophages”, are activated by granulocyte-macrophage colony-stimulating factor (GM-CSF), Toll-like receptor (TLR) ligands such as LPS, or Th1-type cytokines such as IFN-γ [15,30,31,32]. These macrophages are characterized by a high expression of proinflammatory cytokines and high antigen presentation, which play a critical role in promoting inflammation, killing microorganisms, and fighting against tumors [6,33,34]. M1 macrophages release a large number of proinflammatory cytokines, including tumor necrosis factor-α (TNF-α), IL-1β, IL-6, IL-12 and IL-23 [35,36]. Moreover, M1 macrophages highly express CD40, CD64, CD68, CD80, CD86, and human leukocyte DR (HLA-DR) antigen [1,3,26]. In normal tissues, the proportion of M1 macrophages is precisely regulated, and increases during inflammation [29,37]. At the early stage of the inflammatory response, M1 macrophages engulf foreign pathogens and remove bacteria and cell debris. However, excessive secretion of pro-inflammatory cytokines can worsen the inflammatory reaction, resulting in tissue damage and blocked wound healing [3,37].

### 3.2. M2 Macrophages

M2 macrophages, also known as “selectively activated macrophages”, can be activated by macrophage colony-stimulating factor (M-CSF), Th2-type cytokines such as IL-4 and IL-13, immune complexes (ICs), and complement components [7,26]. M2 macrophages play a critical role in anti-inflammatory and immunomodulatory processes and participate in tissue repair and remodeling. M2 macrophages secrete anti-inflammatory cytokines, including transforming growth factor-β (TGF-β), IL-10, and chemokines CCL18 and CCL22 [31]. Generally, M2 macrophages highly express CD163, CD206, CD209, scavenger receptor A (SR-A), SR-B1, CCR2, CXCR1, and CXCR2 [29,38]. According to different microenvironments and stimuli, M2 macrophages can be further divided into four different subtypes: M2a, M2b, M2c, and M2d [31,39]. M2a macrophages are produced by non-polarized macrophages in response to IL-4 and IL-13 stimulation and are associated with allergic reactions. M2a macrophages express high levels of CD163, CD206, IL-1R, Fizz1, Arg1, and Ym1/2, and secrete profibrotic factors required for tissue repair such as TGF-β, insulin-like growth factor (IGF), fibronectin, CCL13, CCL17, CCL18, CCL22, and CCL24, and collect eosinophils, basophils, and Th2 cells [3,7,40,41,42]. M2b macrophages are induced by TLRs, ICs, and IL-1R, and participate in anti-inflammatory responses and immune regulation. M2b macrophages are characterized by a high expression of CD14 and CD80 and secrete anti-inflammatory cytokines such as IL-10. They also recruit regulatory T cells, which help fight inflammation [43,44,45]. M2c macrophages, induced by IL-10, TGF-β, and glucocorticoid (GC), play a key role in inhibiting immune responses and promoting tissue repair. These macrophages express high levels of CD163, CD206, and advanced glycation end product receptors (RAGE) on their surfaces, and secrete high levels of IL-10, TGF-β, CCL16, and CCL18, and recruit eosinophils and resting T cells [41,42,46,47]. M2d macrophages, also known as tumor-associated macrophages (TAMs), are induced by adenosine A2 receptor (A_2A_R), leukemia inhibitory factor (LIF) and IL-6, and mainly inhibit inflammatory reaction and promote angiogenesis and tumor growth [48,49]. These macrophages express high levels of CD163 and vascular endothelial growth factor (VEGF), and secrete significant amounts of IL-10, TGF-β, CXCL10, CXCL16, and CCL5 [3,50].

### 3.3. Signaling Pathways Regulating Macrophage Polarization

Macrophage polarization is a complex process involving multi-factor interactions and is regulated by multiple intracellular signaling molecules and pathways. In recent years, researchers have been studying these pathways in-depth. There are five well-defined signaling pathways involved in macrophage polarization: the PI3K/AKT signaling pathway, TLRs/NF-κB signaling pathway, JAK-STAT signaling pathway, Notch signaling pathway, and JNK signaling pathway [51,52] (Figure 3). In different microenvironments, these pathways individually or collectively regulate macrophage polarization to achieve a dynamic balance between M1 and M2 macrophages.

#### 3.3.1. PI3K/AKT Signaling Pathway

The PI3K/AKT signaling pathway is crucial for cell growth, development, migration, and macrophage polarization [53]. PI3K is a phosphatidylinositol kinase that phosphorylates phosphatidylinositol bisphosphate (PIP2) to produce phosphatidylinositol trisphosphate (PIP3). Cell signaling factors activate class I PI3K, which converts the second messenger PIP2 to PIP3. PIP3 binds to and activates AKT. The activated AKT then regulates cellular functions by regulating the transcription of downstream target genes. The PI3K/Akt1 pathway inhibits TRAF6 and upregulates IRAK-M, the TLR4 repressor, to inactive transcription factor FOXO1, which promotes M2 macrophage polarization by inhibiting TLR4 target genes [54]. Conversely, inhibiting PI3K enhances NF-κB activation and iNOS synthesis, promoting M1 macrophage polarization. Inhibition of the negative regulator of PI3K leads to a decrease in macrophage pro-inflammatory cytokine secretion and induces the synthesis of M2 macrophage surface markers, predisposing macrophage phenotype to M2 [54,55]. Studies have shown that Akt1 and Akt2 have different effects on macrophage polarization. Akt1-deficient macrophages expressed high levels of iNOS, TNF-α, and IL-6. Akt2-deficient macrophages expressed high levels of Fizz1 and IL-10 [53]. Akt1 promotes M2 macrophage polarization and inhibits M1 macrophage polarization. In contrast, Akt2 promotes M1 macrophages and inhibits M2 macrophages.

#### 3.3.2. TLRs/NF-κB Signaling Pathway

In the resting state, NF-κB binds to IκB in an inactive form in the cytoplasm. Upon phosphorylation of IκB, it dissociates from NF-κB, allowing free NF-κB to enter the nucleus and bind to the gene regulatory regions, thereby regulating gene expression [56]. The TLRs/NF-κB signaling pathway plays an important role in macrophage polarization, with all TLRs except TLR3 being able to activate and dissociate NF-κB. TLRs binding to specific ligands, such as PAMP and cytokines, initiate NF-κB-mediated signaling processes that increase the expression of pro-inflammatory-related genes, predisposing the macrophage phenotype towards M1 [57]. The first TLR identified was TLR4, and NF-κB is a key transcriptional regulator of TLR4-induced polarization of macrophages toward the M1. TLR4 activates kinases through both MyD88-dependent and MyD88-independent pathways to phosphorylate IκB, promoting the activation of NF-κB. This leads to the expression of M1 macrophage-related genes by binding to specific DNA sequences in the nucleus [58]. Inhibition of NF-κB reduces TNF-α, IL-1β, and IL-6 in the LPS-induced lung injury model, thereby inhibiting M1 macrophage polarization [59].

#### 3.3.3. JAK-STAT Signaling Pathway

The JAK/STAT signaling pathway is widely involved in important physiological processes, including cell development and immune regulation [60,61]. This pathway involves four kinds of JAK (JAK1, JAK2, JAK3, and TYK2) and seven transcription factors STAT (STAT1, STAT2, STAT3, STAT4, STAT5a, STAT5b, and STAT6). STAT protein family members are key transcription factors regulating M1/M2 polarization of macrophages [60]. When stimulated by IFN-γ, STAT1 in the JAK/STAT signaling pathway is activated, leading to the secretion of pro-inflammatory factors and promoting the polarization of M1 macrophages [62]. IFN-α and IFN-β inhibit STAT1 phosphorylation and M1 macrophage polarization through negative feedback regulation [63]. IL-4 is a key cytokine promoting the polarization of macrophages towards M2 phenotype by activating the STAT3 signaling pathway [64]. In addition, IL-4 binds to its receptor to activate JAK, which on the one hand directly mediates STAT6 phosphorylation to induce M2 macrophage polarization. Meanwhile, phosphorylated STAT6 binds to KLF4 and PPAR-γ to promote M2 macrophage polarization [65]. Suppressors of cytokine signaling (SOCS) are inhibitors of the JAK/STAT signaling pathway. SOCS1-7 directly inhibit JAK activity by binding to JAK or regulate macrophage polarization by inhibiting STAT phosphorylation with competing receptor binding sites [66]. SOCS1 is an endogenous inhibitor of the STAT1 pathway, inhibiting the JAK/STAT1 signaling pathway through its KIR and SH-2 domains and promoting M2 macrophage polarization. As a negative regulator of STAT3, SOCS3 up-regulation can inhibit M2 macrophage polarization [67]. At present, there are few studies on the regulation of macrophage polarization by the JAK/STAT2/4/5 pathway, and the mechanism requires further exploration.

#### 3.3.4. Notch Signaling Pathway

The Notch signaling pathway has complex and diverse functions that are involved in important physiological processes such as immune cell development, angiogenesis, and embryonic development, and are closely related to tumor formation and some neurological diseases. There are four kinds of Notch receptors: Notch1, Notch2, Notch3, and Notch4, which are expressed in various tissues and organs. Notch ligands are classified into two classes: Delta-like and Jagged, with the former consisting of Dll1, Dll3, and Dll4, and the latter consisting of Jagged1 and Jagged2 [68]. The interaction of the Notch receptor and Notch ligand results in the cascade activation of Notch signaling. After the Notch receptor is activated, its intracellular domain (NICD) translocates from the medial cell membrane into the nucleus, activating the transcription of promoter containing the RBP-J recognition site to activate the Notch signaling pathway under the action of the depolymerase and metalloproteinase domain (ADAM)-type protein and γ-secretase complex [69,70]. Studies have shown that the Dll4/Notch1 signaling axis plays an essential role in promoting M1 polarization of macrophages, while blocking the M2 polarization of macrophages and the expression of related cytokines [71]. The specific mechanism is that ligand Dll4 binds Notch1 receptor to form a complex to activate downstream ADAM protease and γ-secretase, leading to NICD entering the nucleus and interacting with RBP-J to promote the polarization of M1 macrophages. In the high-fat diet-induced obesity model, the activation of the Notch1 signaling pathway promotes the M1 polarization of adipose tissue macrophages [72]. Recent studies have shown that NICD can also directly bind to NF-κB protein, acting independently of RBP-J, namely the non-classical Notch signaling pathway [73].

#### 3.3.5. JNK Signaling Pathway

The MAPK pathway is a conserved tertiary kinase cascade, consisting of MKKK, MKK, and MAPK [74]. The JNK signaling pathway, an important branch of the MAPK pathway, is mainly involved in the functional transformation of adipose tissue macrophages. In obesity, growth factors, cytokines, and saturated fatty acids can activate JNK, leading to the transformation of macrophages into M1 phenotype. At the same time, macrophages also activate the M2 macrophage transcription factor SMAD3, which limits the formation of M1 macrophages [75]. LPS upregulates CCL-2 expression through the TLR4/MyD88 signaling pathway, thereby activating JNK and transcription factors of NF-κB/AP-1, which promote M1 macrophages polarization [76,77]. Furthermore, in IL-4 activated macrophages, K63 ubiquitination of MSR1 leads to JNK activation, promoting the transition from an anti-inflammatory to a pro-inflammatory state and promoting M1 polarization of macrophages [78].

## 4. The Role of Macrophage Polarization in Lung Disease

### 4.1. Macrophages in Lung Tissues

Lung tissue macrophages constitute the most abundant immune cell population in the lungs and play a pivotal role in regulating the local microenvironment and immune response [79,80]. They are widely distributed throughout the lung and alveolar tissues and participate in almost all physiological and pathological processes of the lungs [26,81]. There are three main types of lung macrophages: alveolar macrophages (AMs), interstitial macrophages (IMs), and bronchial macrophages (BMs), of which AMs account for over 90% [82]. BMs, isolated from induced sputum, have been less studied [83]. IMs, located in the mesenchyme, interact with mesenchymal lymphocytes to enhance specific immune responses, typically via antigen-presenting macrophages [80,84]. AMs reside in the alveolar lumen and play a crucial role in maintaining lung homeostasis and immune regulation [85,86,87]. Based on their functional status and origin, AMs can be further divided into two subgroups: long-term resident AMs and recruited AMs. Long-lived resident AMs are a uniform, stationary, and immunosuppressed population, whereas recruited AMs are formed when peripheral blood mononuclear cells are recruited into the alveolar lumen in response to certain stimuli [14]. AMs are extremely plastic and heterogeneous. On the one hand, this property drives their rapid response to stimuli and targeted specific immune functions. On the other hand, this property promotes AMs polarization into functionally distinct phenotypes, with different cytokine secretion profiles that mediate pro-inflammatory or anti-inflammatory responses [26,88]. Most studies have focused on AMs in humans and mice.

### 4.2. Macrophage Polarization and Lung Disease

Recently, studies have highlighted the involvement of macrophages in the pathophysiology of various lung diseases, including ALI, COPD, asthma, tuberculosis, and lung cancer (Figure 4). Modulating macrophage phenotypes can regulate the immune homeostasis of these diseases, providing a novel strategy for their treatment. In vitro and in vivo studies have confirmed that many drugs capable of regulating macrophage polarization exhibit potential in treating lung diseases (Table 1).

#### 4.2.1. Macrophage Polarization and ALI

ALI is a manifestation of the systemic inflammatory response syndrome that affects the lungs, and is characterized by an acute inflammatory response in the alveoli and lung parenchyma [14]. Various infectious and non-infectious factors can trigger ALI, such as bacteria, viruses, pneumonia, aspiration, and trauma [107]. Under the influence of these inducers, macrophages and inflammatory cells infiltrate the lung tissues and release numerous cytokines and inflammatory mediators. This leads to the impairment of the integrity of alveolar epithelial cells and pulmonary microvascular endothelial cells, resulting in pulmonary edema. Then, pulmonary edema causes impaired gas exchange and hypoxemia. Acute respiratory distress syndrome (ARDS) may occur if the condition worsens [108]. Macrophages in lung tissue play an important role in all stages of ALI, with AMs being especially prominent as the largest proportion of macrophages [109].

During ALI exudation, a large number of inflammatory mediators and inflammatory cells accumulate in the lungs, causing diffuse alveolar injury and increased permeability of pulmonary capillaries. This results in protein exudation, pulmonary tissue edema, and damage to epithelial barrier and capillary endothelium [14]. M1 macrophages play a pivotal role in the initiation of ALI exudation. During this process, TLRs or other recognition receptors are induced and activated, leading to the polarization of alveolar macrophages towards M1 macrophages and the release of various inflammatory factors, such as IL-1β, IL-6, IL-12, MCP-1, MIP-2, TNF-α, and ROS [110]. Ultimately, this leads to diffuse alveolar injury. Macrophages polarize to the M1 phenotype by activating the classical JAK/STAT1 signaling pathway, where IFN-γ binding to IFN-γR on the surface of macrophages activates signal molecules such as JAK1, JAK2, and STAT1. This promotes the release of inflammatory cytokines, causing macrophages to polarize to M1 [66,111]. SOCS, mainly SOCS1 and SOCS3, negatively regulate the JAK/STAT1 pathway [112].

Following the exudation phase, the convalescence phase is the second stage of ALI. During this phase, a new extracellular matrix is produced in the alveoli, which is accompanied by neovascularization, thereby promoting repair of the injured lung tissue. M2 macrophages play a key role in the regulation of ALI recovery and tissue repair. These macrophages are capable of enhancing the expression of IL-10, fibronectin 1, and TGF-β, while inhibiting the expression of pro-inflammatory cytokines and slowing down the injury of epithelial cells. This enables the promotion of lung tissue repair [113,114]. In addition, a large number of M2 macrophages can further release anti-inflammatory factors, such as IL-4 and IL-10 through phagocytosis of apoptotic neutrophils, thus inhibiting inflammation and facilitating the recovery from a lung injury [115,116]. During ALI recovery, the transformation of macrophages from M1 phenotype to M2 is also regulated by several pathways, including the JAK/STAT and IRF signaling pathways [117].

Pulmonary fibrosis is the advanced stage of the ALI pathologic processes, characterized by the proliferation of fibroblasts and excessive extracellular matrix (ECM) deposition [117]. At this stage, M1 and M2 macrophages are recruited to the site of the lung tissue injury to regulate the fibrosis process, following the destruction of the basement membrane. M1 macrophages play a significant role in matrix degradation through direct and indirect production of MMP and a variety of anti-fibrotic cytokines [118,119]. MMP production is particularly important for ECM remodeling, and it helps to reduce the pathological fibroplasia observed in the late stage of ALI. Interestingly, M2 macrophages promote fibroplasia in lung tissue. They express significant levels of anti-inflammatory cytokines and tissue inhibitors of metalloproteinases, promoting ECM deposition in the lung tissue [14,120]. Therefore, the degree of pulmonary fibrosis depends on the balance between M1 and M2 macrophages in the local microenvironment of the lung tissue injury.

Macrophages are critical immune cells involved in all stages of ALI. Currently, drugs such as dexamethasone, prednisone, and ulinastatin are commonly used in clinical treatments for ALI [121]. In recent years, researchers have focused on developing novel drugs that can regulate macrophage polarization for ALI treatment. Several studies have shown that drugs such as 5-methoxyflavone, zanubrutinib, canagliflozin, and κ-Opioid receptor agonist U50448H can inhibit macrophages with the M1 phenotype and promote macrophages with the M2 phenotype, which provides a potential candidate drug for the treatment of ALI [89,90,91,92].

#### 4.2.2. Macrophage Polarization and COPD

COPD is an inflammatory disease of the lungs that affects both the lung parenchyma and airways, resulting in the obstruction of small airways and emphysema [13]. This disease is characterized by progressive airflow restriction and recurrent inflammation, massive mucus secretion in the central airway, fibrosis of the small airway, and destruction of lung parenchyma. Smoking is the primary cause of COPD, which induces the activation and aggregation of macrophages [122]. These induced and activated macrophages play a crucial role in the pathogenesis of COPD by participating in inflammation, tissue repair, phagocytosis, and regulating the pulmonary microenvironment.

In COPD, inflammatory responses affect bronchi, bronchioles, and lung parenchyma, resulting in progressive airflow restriction and intrapulmonary hypoxia. At the early stage, M1 macrophages release pro-inflammatory mediators, such as IL-1, TNF-α, NO, ROS, CCL2, and CXCL1, to stimulate the adaptive immune response and remove foreign irritants [123]. In the middle and late stages, M2 macrophages become dominate, inhibiting excessive inflammatory responses and maintaining homeostasis [124]. In addition, M2 macrophages have the function of tissue repair and can secrete a variety of anti-inflammatory factors in the lung tissue of COPD patients [125]. Chronic lung inflammation due to long-term cigarette smoke exposure can cause macrophages to overexpress CD206, IL-10, and TGF-β, driving them to polarize into M2 phenotype and promote tissue repair, as studies have shown [126].

Phagocytosis is a critical function of macrophages in maintaining immune function. M2 macrophages are the primary cells responsible for pulmonary phagocytosis and clearance in COPD. Studies have confirmed that reduced phagocytosis capacity of pulmonary macrophages in COPD patients is associated with a decrease in key markers of M2 macrophage, such as CD206 and CD163, indicating the importance of M2 macrophages in phagocytosis [127]. Additionally, as COPD progresses, the number of M2 macrophages decreases, reducing their ability to phagocytose the accumulated debris from earlier stages of the disease, resulting in the weaker phagocytic activity of macrophages in the middle and late stages of COPD.

Macrophages exhibit tissue-specific characteristics, and their polarization phenotypes vary across different tissues. The most obvious example is the difference in macrophage phenotype between the airway and alveolar lumen in patients with COPD. In the airway, pro-inflammatory M1 macrophages dominate, while anti-inflammatory M2 macrophages are predominant in the alveolar lumen [128]. Moreover, M2 macrophages accounted for the majority of sputum and bronchial alveolar lavage fluid in patients with COPD [13,123]. COPD is a chronic, progressive disease and different pulmonary microenvironments in its progressive stage can alter the polarization of macrophages. Clinical studies have reported a reduction in the proportion of anti-inflammatory phenotype M2a macrophages in the lung tissues of COPD patients, but this proportion returns to a normal amount after a year of treatment [129].

Currently, the treatment of COPD relies mainly on the use of bronchodilators, anti-inflammatory drugs, and anti-infective drugs [13]. While these medications have significant therapeutic effects, they can also cause side effects. Changes in the lung microenvironment can induce different phenotypes of macrophages, resulting in different functional properties. Therefore, modulating the macrophage phenotype in COPD is a novel approach to treat COPD. Ulinastatin, a drug that inhibits M1 macrophages, has been shown to protect the lungs of mice with COPD and may be a potential candidate drug for the treatment of COPD [93].

#### 4.2.3. Macrophage Polarization and Asthma

Asthma is a heterogeneous chronic lung disease characterized by airway inflammation, reversible airflow obstruction, airway remodeling, and bronchial hyperresponsiveness [130]. Allergic asthma, which is caused by allergic stimulation, is the most common type, while non-allergic asthma, caused by non-allergic stimulation, is another common form. Allergens trigger bronchoconstriction and increase the number of eosinophils in the lungs in allergic asthma, while non-allergic asthma usually occurs in adults with a neutrophilic lung infiltration [131]. Although eosinophils and neutrophils are considered as key factors in asthma pathogenesis, recent studies have found that the macrophage phenotypes change in asthma, suggesting an important role for macrophages in asthma.

Macrophages are abundant in lung tissue, but their role in asthma pathology appears to be more related to functional changes than quantitative differences. There is no significant difference in the percentage of macrophages in lung tissue and bronchoalveolar lavage fluid between asthmatic patients and control groups [132]. The polarization and phenotype of macrophages are usually related to the type of asthma. Robbe et al. replicated an allergic asthma mouse model and a non-allergic asthma mouse model to identify M1 or M2 macrophages in different types of asthma [133]. The results showed that macrophages had M1 polarization in the non-allergic asthma mouse model, while macrophages had M2 polarization in the allergic asthma mouse model. This indicates that M1 macrophages are the primary effector cells in non-allergic asthma, while M2 macrophages are the dominant effector cells in allergic asthma. M1 macrophages cause asthma airway inflammation by producing pro-inflammatory cytokines such as TNF-α and IL-6 [134]. In addition, M1 macrophages have been implicated in the pathophysiology of severe asthma, especially in patients who do not respond well to corticosteroids [21,135]. In the pathogenesis of asthma, M2 macrophages promote allergic inflammation and airway hyperreactivity, and through collagen deposition, they induce airway remodeling, leading to asthma exacerbation [12].

Reducing inflammation is a crucial strategy for treating asthma. Corticosteroids are commonly used in clinical practice to inhibit the production of proinflammatory cytokines. Targeting the NLRP3 inflammasome is a potential strategy for treating asthma. Inhibition of NLRP3 inflammasome can prevent airway hyperreactivity and inflammation in mice with severe asthma by inhibiting the expression of IL-1β and Th-2 [136]. Recently, some studies have proposed a novel strategy for suppressing the inflammatory response in asthma by regulating macrophage polarization. Azithromycin has been successfully used for the treatment of asthma by promoting a shift in macrophages from the M1 phenotype to the M2 [94]. Furthermore, protostemonine, emodin, and tiotropium have been found to inhibit the polarization of M2 macrophages and reduce inflammation in asthma patients [95,96,97]. Obviously, regulating macrophage polarization holds great promise as a treatment for asthma. In allergic asthma, M1 macrophages should be inhibited, while in non-allergic asthma, M2 macrophages should be restricted.

#### 4.2.4. Macrophage Polarization and Tuberculosis

Tuberculosis (TB) is a chronic infectious disease caused by the bacterium mycobacterium tuberculosis (MTB), which can infect organs throughout the body, most commonly the lungs. A key histological feature of TB is the tuberculous granuloma [137].

MTB primarily infects macrophages, and macrophage polarization plays a crucial role in MTB infection and the formation and development of tuberculous granulomas [138]. M1 and M2 macrophages coexist in tuberculous granuloma, but M1 macrophages are dominant in the early stages of infection and gradually converted to M2 macrophages in the later stages. Therefore, M1 macrophages are considered the primary cells promoting the formation of granuloma, while M2 macrophages inhibit granuloma formation [11]. In rat models, after injection with Bacillus Calmette–Guérin, a large number of M1 macrophages were detected at the peak of liquefaction necrosis in tuberculous lesions, while M2 macrophages were rare [139].

MTB secretes proteins that can affect macrophage polarization. For example, tubercule-specific peptides, such as E6, E7, and C14 in ESAT-6 and CFP-10 secreted by MTB have varying effects on macrophage polarization [140,141]. E6 and C14 promote M2 polarization of macrophages, while E7 has a bidirectional regulatory effect on macrophage polarization. In the early stage, E7 induces macrophages to M1 polarization through the JNK pathway, but in the later stage, it promotes M2 polarization via the JAK/STAT pathway. In addition, the heat shock protein HSP16.3 of MTB has been found to promote the polarization of macrophages from M1 to M2 and inhibit the development of inflammation [142].

The frontline treatment of MTB infection is a combination of four drugs: isoniazid, rifampicin, pyrazinamide, and ethambutol. These drugs aid in the polarization of M1 macrophages to the M2 phenotype [98]. However, the challenge of treating TB is further complicated by the emergence of multidrug-resistant (MDR) and extensively drug-resistant (XDR) TB. Thus, there is an urgent need to develop new anti-TB drugs and combination treatment regimens [143]. At present, most of the studies on macrophage polarization in TB have focused on the structure and function of MTB. To screen new therapeutic drugs for TB that target macrophage polarization, further research is needed.

#### 4.2.5. Macrophage Polarization and Lung Cancer

Lung cancer has the highest mortality rate among all types of cancer worldwide, with an average five-year survival rate of only about 20% [144]. It is mainly divided into two types: small cell lung cancer (SCLC) and non-small cell lung cancer (NSCLC), with NSCLC accounting for about 80% of all lung cancer cases [10]. In the past, the treatment of lung cancer mainly focused on tumor cells. However, with the development of medical research, the tumor microenvironment (TME) in lung cancer has gradually gained attention. Tumor-associated macrophages (TAMs) are macrophages that infiltrate or aggregate in TME and can affect the entire process of lung cancer development.

TAMs are highly malleable, able to polarize to different phenotypes and perform different functions depending on the microenvironmental stimuli. Studies have shown that M1 TAMs are dominant in the early stage of lung cancer, while M2 TAMs become dominant in the middle and late stages [145]. M2 TAMs play crucial roles in the growth, angiogenesis, metastasis, and invasion of lung cancer. Severe infiltration of M2 TAMs is associated with a worse prognosis and lower survival rates in patients. Oct4 expression in lung cancer cells has been shown to promote the polarization of macrophages towards M2 TAMs by upregulating M-CSF, thus leading to tumor growth and metastasis [146]. In mouse models of lung cancer, hypoxia has been found to increase M2 TAMs and promote the expression of IL-10, VEGF, and HIF-1α, further promoting macrophage invasion and cancer cell metastasis [147]. Knockout of IFN-γ in a mouse model of lung cancer resulted in TAMs polarizing into the M2 phenotype and produced larger lung tumors than control mice [148]. In addition, M2 TAMs may promote the expression of EMT and CRYAB, which in turn promote the invasion and metastasis of lung cancer cells [149].

M2 TAMs play a critical role in promoting tumor angiogenesis and local tumor invasion. Studies have demonstrated that M2 TAMs can increase the expression of VEGF-A and VEGF-C in NSCLC cells, leading to the formation of blood vessels and lymphatic vessels at the tumor site [150]. G-Rh2 has been found to induce M2 TAMs to differentiate into M1 phenotype, thereby inhibiting the expression of angiogenic factors and the migration of lung cancer cells [151]. In addition, M2 TAMs also inhibited tumor immunity. M2 TAMs stimulated by Th2 cells release a variety of cytokines, enzymes, and chemokines in the lung TME, which recruit regulatory T cells or inhibit cytotoxic T cell activity, leading to immune tolerance [144,152]. ILT4 is induced by activation of the EGFR-AKT and ERK1/2 signaling pathways in NSCLC cells. Overexpression of ILT4 has been found to recruit M2 TAMs that impair T cell response and inhibit tumor immunity, while inhibition of ILT4 can prevent immunosuppression and inhibit tumor growth [153].

Given the critical role of TAMs in the occurrence and development of lung cancer, targeting TAMs with therapeutic strategies may enhance the efficacy of systemic chemotherapy and immunotherapy for lung cancer. Currently, three strategies are being considered to inhibit the development of lung cancer: changing the phenotype of TAMs, preventing the recruitment of TAMs, and depleting TAMs. Studies have shown that CD40 is involved in the differentiation of monocytes into M1 TAMs and can reverse M2 TAMs into M1. CD40-targeted agonists are being investigated in combination with chemotherapy or immune-checkpoint inhibitors in preliminary clinical trials for NSCLC [154]. TAMs receptor inhibitors are another therapeutic approach to reversing the transformation of M2 TAMs into M1. TAMs receptors, including Tyro3, Axl, and MERTK, polarize TAMs toward the M2 phenotype [155]. In mouse NSCLC models, studies have demonstrated that MERTK’s small molecule inhibitor UNC2025 can inhibit tumor metastasis [156].

Chemokines, cytokines, and complement mediators contribute to the recruitment of TAMs in the tumor microenvironment. Therefore, regulating these factors to inhibit TAM recruitment is an interesting research direction. At present, the CCL2-CCR2 and CSF-1-CSF-1R signaling pathways have been studied extensively [157,158]. The CCL2-CCR2 signaling pathway is known to recruit TAMs and promote tumor neovascularization and the rapid growth of cancer cells. In a mouse model of lung cancer, it was found that knockout of the CCR2 gene or use of the CCR2 inhibitors inhibited the recruitment of TAMs and promoted their polarization into M1 phenotype, ultimately inhibiting the progression of lung cancer [159].

The number of TAMs in the tumor tissue is closely related to the prognosis of the disease, and higher numbers are associated with faster tumor growth. Therefore, consumption of TAMs in TME is an effective strategy for inhibiting tumor growth. CSF-1R is expressed in TAMs, and CSF-1R inhibitors can deplete M2 TAMs in TME. Currently, inhibitors of CSF-1R monoclonal antibodies combined with immunotherapy drugs are being tested in clinical trials for NSCLC treatment [160]. Targeting cell surface molecules, such as CD52, SR-A, folicacid receptor-β (FR-β), and CD206 can also lead to the depletion of M2 TAMs, thereby inhibiting angiogenesis and delaying tumor progression [161].

Taken together, targeting TAMs is an innovative therapeutic for lung cancer treatment. While significant progress has been made in developing novel anti-tumoral drugs that target TAMs, the clinical benefits of some of these drugs remain to be determined [162]. Several studies have demonstrated the potential of drugs such as imatinib, β-elemene, resveratrol, paeoniflorin, paeoniflorin, hydroxychloroquine, astragaloside IV, puerarin, and gefitinib for lung cancer treatment by inhibiting the polarization of M2 TAMs or transforming M2 TAMs into M1 [99,100,101,102,103,104,105,106].

## 5. Conclusions

Macrophages are the major population of immune cells and play a crucial role in pathogen recognition, clearance, and innate and acquired immune responses. They can be polarized into M1 and M2 phenotypes depending on the local microenvironment. In this review, we focused on the origin of macrophages, the phenotype and polarization of macrophages, as well as the signaling pathways involved in macrophage polarization. We also highlighted the role of macrophage polarization in lung diseases. Many drugs that can regulate the polarization of macrophages have the potential to treat lung diseases. Thus, we believe targeting and modulating macrophage phenotypes is an effective and promising strategy for the treatment of lung diseases.

## Figures and Tables

**Figure 1 molecules-28-02379-f001:**
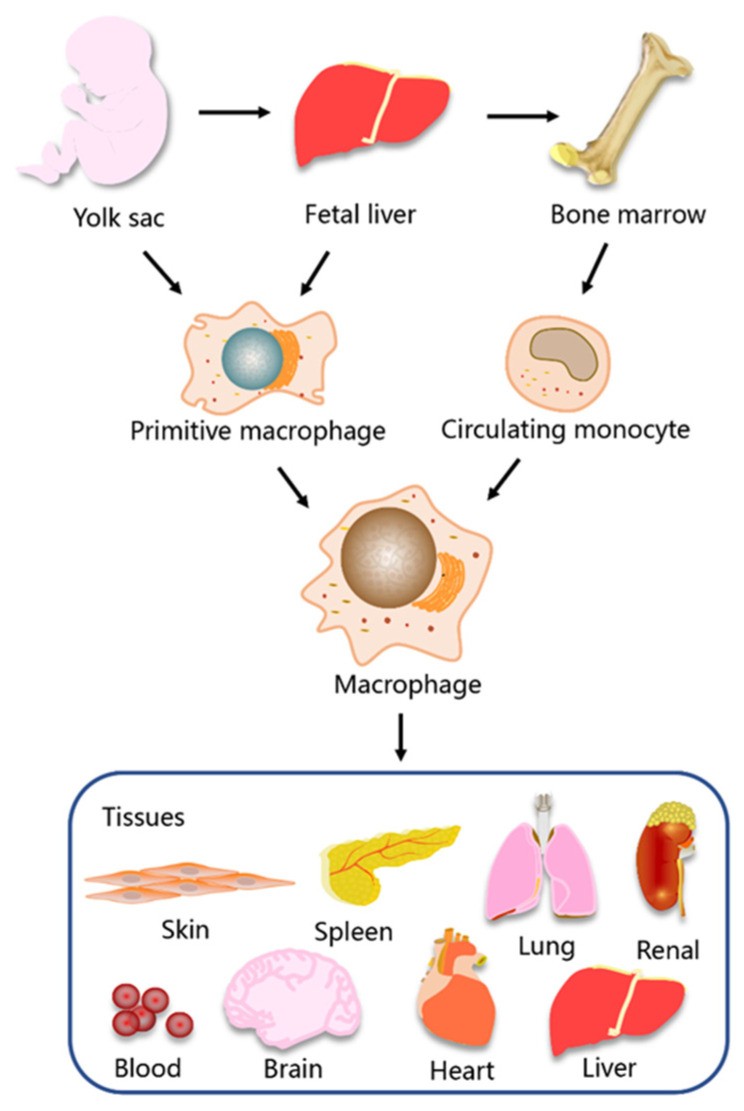
The origin and tissue distribution of macrophages. Primitive macrophages are derived from the yolk sac and fetal liver and circulating monocytes are derived from bone marrow. Macrophages are widely distributed in tissues and organs of the body, including blood, skin, brain, heart, lung, liver, spleen, and kidney.

**Figure 2 molecules-28-02379-f002:**
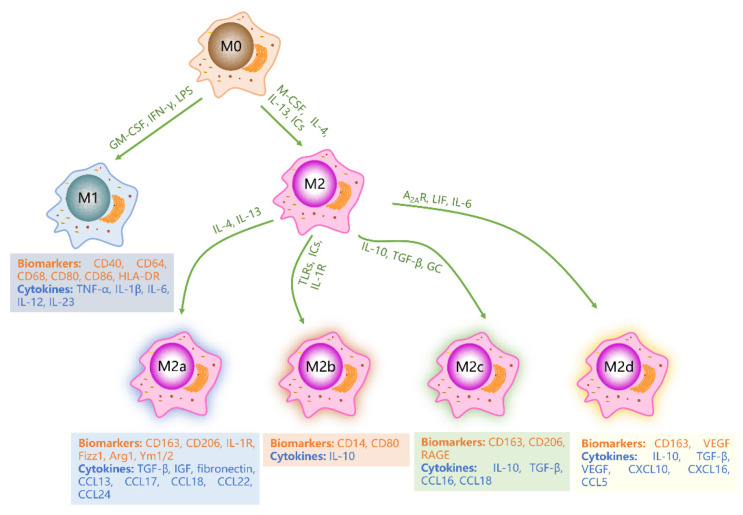
Schematic diagram of macrophage subtypes. Under stimulation by GM-CSF, IFN-γ, and LPS, M0 macrophages polarize into M1 macrophages. Alternatively, M-CSF, IL-4, IL-13, and IC stimulation lead to polarization of M0 macrophages into M2 macrophages. Various cytokines further induce M2 macrophages to differentiate into M2a, M2b, M2c, and M2d phenotypes. M1, M2a, M2b, M2c and M2d macrophages express distinct biomarkers and secrete diverse cytokines. Green letters represent cytokines or factors that promote macrophage polarization. Orange letters represent biomarkers of macrophages. Blue letters indicate cytokines secreted by macrophages.

**Figure 3 molecules-28-02379-f003:**
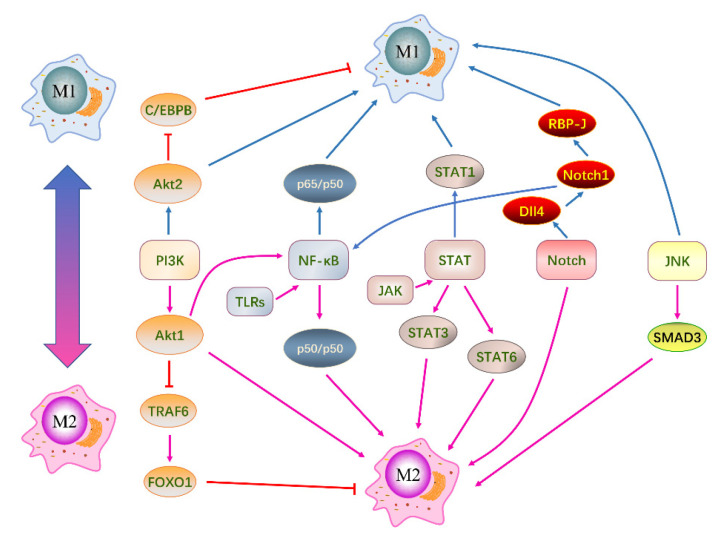
Signaling pathways associated with macrophage polarization. There are five primary signaling pathways: PI3K/AKT signaling pathway, TLRs/NF-κB signaling pathway, JAK-STAT signaling pathway, Notch signaling pathway, and JNK signaling pathway. These pathways modulate macrophage polarization individually or in conjunction with others. Negative regulation between adjacent molecules is indicated by red lines, while positive regulation is represented by arrows. Molecules that promote macrophage polarization toward the M1 phenotype are depicted by blue arrows, and those that promote M2 polarization are shown by pink arrows.

**Figure 4 molecules-28-02379-f004:**
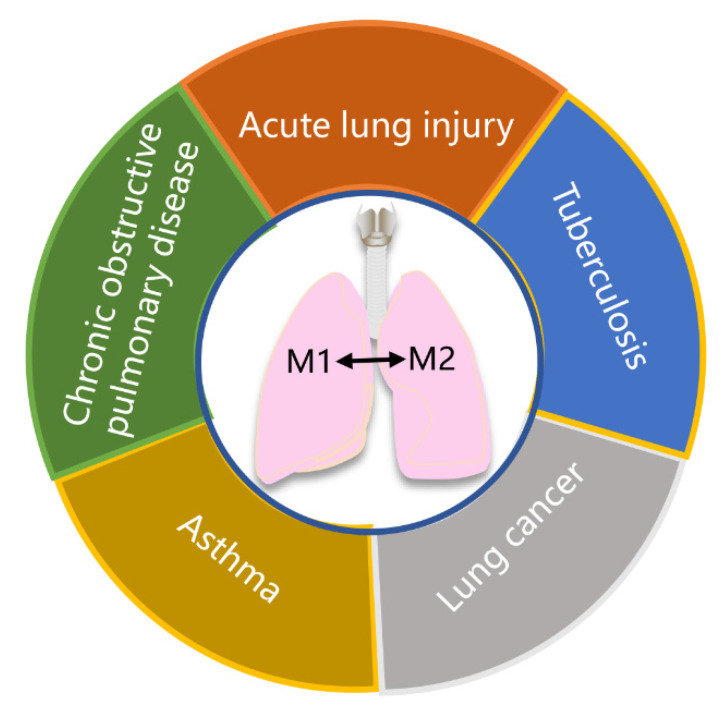
Schematic diagram of important lung diseases involving macrophage polarization. Macrophage polarization plays a crucial role in the pathophysiology of various lung diseases, including acute lung injury, chronic obstructive pulmonary disease, asthma, tuberculosis, and lung cancer.

**Table 1 molecules-28-02379-t001:** Major phenotypes of macrophages in lung diseases and potential therapeutic agents.

Names of Diseases	Major Phenotypes of Macrophages	Potential Therapeutic Agents and Their Effects	Reference
Acute lung injury	Exudation phase: M1 macrophages. Convalescence phase: M2 macrophages. Pulmonary fibrosis: M1 and M2 macrophages.	5-methoxyflavone: inhibiting the polarization of M1 macrophages and the repolarization of M2 macrophages to the M1 phenotype.	[89]
		Zanubrutinib: inhibiting M1 macrophage polarization and promoting the polarization of M2 macrophages.	[90]
		Canagliflozin: inhibiting macrophages with the M1 phenotype and promoting the shift of macrophages from the M1 phenotype to the M2.	[91]
		U50448H: promoting the polarization of M2 macrophages.	[92]
Chronic obstructive pulmonary disease	Early stage: M1 macrophages. Middle and late stages: M2 macrophages.	Ulinastatin: inhibiting M1 macrophage polarization.	[93]
Asthma	Non-allergic asthma: M1 macrophages. Allergic asthma: M2 macrophages.	Azithromycin: promoting macrophage polarization from M1 phenotype to M2.	[94]
		Protostemonine: inhibiting M2 macrophage polarization.	[95]
		Emodin: Suppressing M2 macrophage polarization.	[96]
		Tiotropium: inhibiting M2 macrophages.	[97]
Tuberculosis	Early stage: M1 macrophages. Late stage: M2 macrophages.	Combination of isoniazid, rifampicin, pyrazinamide and ethambutol: promoting M1 macrophages to shift to M2 phenotype.	[98]
Lung cancer	Early stage: M1 tumor-associated macrophages (TAMs). Middle and late stages: M2 TAMs.	Imatinib: restraining M2 TAM polarization.	[99]
		β-elemene: converting TAMs from M2 to M1.	[100]
		Resveratrol: suppressing M2 TAMs.	[101]
		Paeoniflorin: inhibiting M2 TAMs.	[102]
		Hydroxychloroquine: transforming M2 TAMs into M1.	[103]
		Astragaloside IV: reducing M2 TAMs.	[104]
		Puerarin: converting M2 TAMs to M1.	[105]
		Gefitinib: inhibiting M2 TAM polarization.	[106]

## Data Availability

Not applicable.

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
