# Peer review of "Macrophage Polarization: An Important Candidate Regulator for Lung Diseases"

_molecules, 2023, doi:10.3390/molecules28052379_

Round 1

Reviewer 1 Report

Comments to the authors:
The authors did a great job reviewing macrophage polarization and their role in lung diseases. I have given some minor comments to make it better.
1. Fig.1 needs to be improved for better quality. The sequential origin of macrophages' yolk sac, fetal liver, and bone marrow can be given. Primitive and circulating macrophage shapes may be differentiated to do that you can use different shapes. You may include all the tissues mentioned as a separate panel and include skin, spleen, and peritoneum.
2. Give a legend for all the figures and briefly describe them
.
3. Wherever possible, graphically represent the macrophage role in diseases. For example, in the ALI section, you may mention the Exudative, proliferative, and fibrotic phages and give the roles of macrophages in each phage so this way you may clearly associate pathology vs. functional roles of macrophages.

4. Fig.3 is confusing, please give specific color arrows for the molecules favoring M1 and M2 and clearly demarcate them.
5. Fig.2, The initial macrophage label as M0. The details given are hard to understand, so make it clear and give space between biomarkers and cytokines. You may tabulate for better representation.

Author Response

Dear Editors and Reviewers:

Thank you for your reviewer for our manuscript entitled “Macrophage polarization: an important candidate regulator for lung diseases (Submission ID: 2233764). And we would like to thank you for your careful reading, helpful comments, and constructive suggestions, which has significantly improved the presentation of our manuscript. We have carefully considered all comments from the reviewers, and revised our manuscript accordingly and marked in red. We believe that our responses have well addressed all concerns from the reviewers. All corrections in the paper and the responds to the reviewer's comments are as following:

1. Fig.1 needs to be improved for better quality. The sequential origin of macrophages' yolk sac, fetal liver, and bone marrow can be given. Primitive and circulating macrophage shapes may be differentiated to do that you can use different shapes. You may include all the tissues mentioned as a separate panel and include skin, spleen, and peritoneum.

Response: Thank you for your advice very much. We have modified Figure 1 to enhance its quality.

2. Give a legend for all the figures and briefly describe them.

Response: Thank you for your advice. We have added legends for all the figures to better describe their content.

3. Wherever possible, graphically represent the macrophage role in diseases. For example, in the ALI section, you may mention the exudative, proliferative, and fibrotic phages and give the roles of macrophages in each phage so this way you may clearly associate pathology vs. functional roles of macrophages.

Response: Thank you for your advice. To better illustrate the role of macrophages in lung diseases, we have added a new table to demonstrate the major macrophage phenotypes in each lung disease and the candidate therapeutic agents in the revised manuscript.

4. Fig.3 is confusing, please give specific color arrows for the molecules favoring M1 and M2 and clearly demarcate them.

Response: Thank you for your advice. We have colored the arrows in Figure 3 to differentiate the molecules favoring M1 and M2.

5. Fig.2, The initial macrophage label as M0. The details given are hard to understand, so make it clear and give space between biomarkers and cytokines. You may tabulate for better representation.

Response: Thank you for your reminding and suggestion. We've labeled the initial macrophage as M0. Additionally, biomarkers and cytokines have been colored to differentiate them.

Reviewer 2 Report

Deng et al. aimed to discuss the origin of macrophages, the phenotype and polarization of macrophages, signaling pathways related to macrophage polarization, and the role of macrophage polarization in lung diseases, contributing to a better understanding of the functions and immunomodulatory features of macrophages.

Although this could be a significant contribution to the field there are major issues that should be resolved before being accepted for publication. 

The phenotype and polarization of macrophages and signaling pathways related to macrophage polarization are redundantly discussed in previous review articles and even textbooks. The most novel part of this review is the role of macrophage polarization in lung diseases which is not well explained. In this regard, the authors should prepare better figures (in terms of quality and information), while figure 4 is not informatic at all, representing the role of macrophage polarization in each lung disease.

The role of macrophages in infectious lung diseases such as pneumonia is underestimated. 

Which drugs are targeting macrophages in each lung disease? Adding a table in this section could help. Even if there are some ongoing clinical trials they could be listed.

What about engineering TAMs in lung cancer? 

Deep English editing is necessary.  

Author Response

Dear Editors and Reviewers:

Thank you for your reviewer for our manuscript entitled “Macrophage polarization: an important candidate regulator for lung diseases (Submission ID: 2233764). And we would like to thank you for your careful reading, helpful comments, and constructive suggestions, which has significantly improved the presentation of our manuscript. We have carefully considered all comments from the reviewers, and revised our manuscript accordingly and marked in red. We believe that our responses have well addressed all concerns from the reviewers. All corrections in the paper and the responds to the reviewer's comments are as following:

1.The phenotype and polarization of macrophages and signaling pathways related to macrophage polarization are redundantly discussed in previous review articles and even textbooks. The most novel part of this review is the role of macrophage polarization in lung diseases which is not well explained. In this regard, the authors should prepare better figures (in terms of quality and information), while figure 4 is not informatic at all, representing the role of macrophage polarization in each lung disease.

Response: Thank you for your advice. We have improved the quality and information provided in the figures and have added a new table to the revised manuscript showing the major macrophage phenotypes in each lung disease. The role of macrophages in lung diseases can be well understood when combined with Figure 2.

2. The role of macrophages in infectious lung diseases such as pneumonia is underestimated.

Response: Thank you for your reminding. We have made a brief reference to infectious lung diseases in the revised manuscript. ALI can be caused by a variety of infectious and non-infectious factors, including bacteria, viruses, pneumonia, aspiration and trauma. We believe that the pathology and mechanism of ALI have included those of infectious lung diseases.

3. Which drugs are targeting macrophages in each lung disease? Adding a table in this section could help. Even if there are some ongoing clinical trials they could be listed.

Response: Thank you for your advice. We have added a new table to the revised manuscript showing the major macrophage phenotypes in each lung disease and the candidate therapeutic agents, including those from ongoing clinical trials.

4. What about engineering TAMs in lung cancer?

Response: Thank you for your reminding. We have added the role of TAMs in the treatment of lung cancer in the revised manuscript and marked it in red.

5. Deep English editing is necessary.

Response: Thank you for your advice very much. We have requested assistance from who are good at English to check and improve the grammar and language of the entire article. We believe that the language of this paper is now suitable for the review process.

Round 2

Reviewer 2 Report

The authors have made extensive improvements to the initial submission and the article could be published after minor grammatical revisions.